# Epidemiology of atrial fibrillation in the *All of Us* Research Program

**Alvaro Alonso**[1]*, **Aniqa B. Alam**[1], **Hooman Kamel**[2], **Vignesh Subbian**[3], **Jun Qian**[4], **Eric Boerwinkle**[5], **Mine Cicek**[6], **Cheryl R. Clark**[7], **Elizabeth G. Cohn**[8], **Kelly A. Gebo**[9], **Roxana Loperena-Cortes**[10], **Kelsey R. Mayo**[10], **Stephen Mockrin**[11], **Lucila Ohno-Machado**[12], **Sheri D. Schully**[13], **Andrea H. Ramirez**[14], **Philip Greenland**[15]

1 Department of Epidemiology, Rollins School of Public Heath, Emory University, Atlanta, GA, United States of America, 2 Clinical and Translational Neuroscience Unit, Department of Neurology and Feil Family Brain and Mind Research Institute, Weill Cornell Medicine, New York, NY, United States of America, 3 Department of Biomedical Engineering, Department of Systems and Industrial Engineering, College of Engineering, BIO5 Institute, The University of Arizona, Tucson, AZ, United States of America, 4 Department of Biomedical Informatics, Vanderbilt University Medical Center, Nashville, TN, United States of America, 5 Human Genetics Center, Department of Epidemiology, Human Genetics, and Environmental Sciences, School of Public Health, The University of Texas Health Science Center at Houston, Houston, TX, United States of America, 6 Department of Laboratory Medicine and Pathology, Mayo Clinic, Rochester, MN, United States of America, 7 Department of Medicine, Brigham and Women's Hospital, Harvard Medical School, Boston, MA, United States of America, 8 Hunter College, City University of New York, New York, NY, United States of America, 9 Department of Medicine, Johns Hopkins University School of Medicine, Baltimore, MD, United States of America, 10 Vanderbilt Institute for Clinical and Translational Research, Vanderbilt University Medical Center, Nashville, TN, United States of America, 11 Life Sciences Division, Leidos, Inc, Frederick, MD, United States of America, 12 Department of Biomedical Informatics, University of California San Diego, San Diego, California, United States of America, 13 *All of Us* Research Program, National Institutes of Health, Bethesda, MD, United States of America, 14 Department of Medicine, Vanderbilt University Medical Center, Nashville, TN, United States of America, 15 Department of Preventive Medicine, Feinberg School of Medicine, Northwestern University, Chicago, IL, United States of America

* alvaro.alonso@emory.edu

## Abstract

### Background

The prevalence, incidence and risk factors of atrial fibrillation (AF) in a large, geographically and ethnically diverse cohort in the United States have not been fully described.

### Methods

We analyzed data from 173,099 participants of the *All of Us* Research Program recruited in the period 2017–2019, with 92,318 of them having electronic health records (EHR) data available, and 35,483 having completed a medical history survey. Presence of AF at baseline was identified from self-report and EHR records. Incident AF was obtained from EHR. Demographic, anthropometric and clinical risk factors were obtained from questionnaires, baseline physical measurements and EHR.

### Results

At enrollment, mean age was 52 years old (range 18–89). Females and males accounted for 61% and 39% respectively. Non-Hispanic Whites accounted for 67% of participants, with

**Data Availability Statement:** Data used for this manuscript is obtained from the All of Us Research Program and not owned by the authors. Data from the All of Us Research Program is accessible only through the Researcher Workbench (https://

workbench.researchallofus.org) as stipulated in the informed consent of participants in the program. This data use agreement prohibits investigators from providing row level data on AllofUs participants and thus providing a de-identified dataset is not possible for this manuscript. The code used for this demonstration project is available within the Researcher Workbench at https://workbench.researchallofus.org/ workspaces/aou-rw-23896673/ afibepidemiologyaouv4/notebooks/preview/Basic. ipynb. Any investigator interested in accessing data used for this manuscript or any other All of Us Research Program data can do so following the procedures outlined in https://www. researchallofus.org/apply/.

**Funding:** The All of Us Research Program is supported by the National Institutes of Health, Office of the Director: Regional Medical Centers: 1 OT2 OD026549; 1 OT2 OD026554; 1 OT2 OD026557; 1 OT2 OD026556; 1 OT2 OD026550; 1 OT2 OD 026552; 1 OT2 OD026553; 1 OT2 OD026548; 1 OT2 OD026551; 1 OT2 OD026555; IAA #: AOD 16037; Federally Qualified Health Centers: HHSN 263201600085U; Data and Research Center: 5 U2C OD023196; Biobank: 1 U24 OD023121; The Participant Center: U24 OD023176; Participant Technology Systems Center: 1 U24 OD023163; Communications and Engagement: 3 OT2 OD023205; 3 OT2 OD023206; and Community Partners: 1 OT2 OD025277; 3 OT2 OD025315; 1 OT2 OD025337; 1 OT2 OD025276. Research reported in this publication was also supported by the American Heart Association award 16EIA26410001 and the National Heart, Lung, And Blood Institute of the National Institutes of Health under Award Number K24HL148521. The content is solely the responsibility of the authors and does not necessarily represent the official views of the National Institutes of Health. The funders had no role in study design, data collection and analysis, decision to publish, or preparation of the manuscript.

**Competing interests:** The authors have declared that no competing interests exist.

non-Hispanic Blacks, non-Hispanic Asians and Hispanics accounting for 26%, 4% and 3% of participants, respectively. Among 92,318 participants with available EHR data, 3,885 (4.2%) had AF at the time of study enrollment, while the corresponding figure among 35,483 with medical history data was 2,084 (5.9%). During a median follow-up of 16 months, 354 new cases of AF were identified among 88,433 eligible participants. Individuals who were older, male, non-Hispanic white, had higher body mass index, or a prior history of heart failure or coronary heart disease had higher prevalence and incidence of AF.

## Conclusion

The epidemiology of AF in the *All of Us* Research Program is similar to that reported in smaller studies with careful phenotyping, highlighting the value of this new resource for the study of AF and, potentially, other cardiovascular diseases.

## Introduction

Atrial fibrillation (AF) is a common cardiac arrhythmia associated with reduced quality of life and increased rates of stroke, heart failure, dementia, and overall mortality [1]. Numerous studies have identified major risk factors for AF, including older age, white race, male sex, hypertension, obesity and other cardiovascular diseases [2]. In the United States, studies on the prevalence, incidence and risk factors of AF have been limited to relatively small but well-phenotyped cohorts [3–5], or larger administrative or clinical databases restricted to individuals enrolled in geographically-restricted healthcare plans [6], receiving care in selected medical institutions [7], enrolled in specific insurance programs [8], or residing in specific regions of the United States [9]. However, no prior studies have described the epidemiology of AF in a large cohort representing the broad diversity of the US population in terms of ancestry, demographic variables, socioeconomic status, geographic location, and other characteristics.

Launched in 2018, the *All of Us* Research Program aims to recruit one million participants in the United States with the primary goal of accelerating biomedical research and improving the health of the public [10]. The program obtains data from participants through self-reported surveys, physical measurements, electronic health records (EHRs), and the collection and analysis of biological samples. Over 80% of study participants are from groups that have been traditionally underrepresented in biomedical research [10]. Still, the reliability of the data being collected, particularly in regards to their ability to identify risk factors for disease status, including AF, has not been established yet.

Therefore, our goal was to characterize the epidemiology of AF in the *All of Us* Research Program to evaluate whether demographic patterns and risk factors of AF established in prior literature replicate in this novel research resource. Replicating these associations would bolster the value of *All of Us* data for future innovative research on AF and other cardiovascular diseases that takes advantage of the diversity of the study population.

## Methods

### The *All of Us* Research Program

The *All of Us* Research Program is a prospective cohort study aiming to recruit at least one million individuals in the United States, with the overall goal of providing a unique resource to study the effects of lifestyle, environment and genomics on health and health outcomes [10]. Participant recruitment is predominantly done through participating health care provider

organizations and in partnership with Federally Qualified Health Centers, with an emphasis on recruiting persons affiliated with those centers. Interested potential participants can also enroll in the program as direct volunteers, visiting community-based enrollment sites. Initial enrollment, informed consent (including consent to share EHRs), and baseline health surveys are done digitally through the *All of Us* program website (https://joinallofus.org). Once this step is completed, the participant is invited to undergo a basic physical exam and biospecimen collection at the affiliated health care site. Participant follow-up is done in two ways, passively via linkage with EHR and actively by periodic follow-up surveys.

For this study, we included data from participants enrolled in the study between May 2017 and August 2019. This was one of several demonstration projects supported by the *All of Us* Research Program. Demonstration projects were designed to describe the cohort, replicate previous findings for data validation, and avoid novel discovery in line with the program value to ensure equal access by researchers to the data [11]. The work described here was proposed by Consortium members, reviewed and overseen by the program's Science Committee, and was confirmed as meeting criteria for non-human subjects research by the *All of Us* Institutional Review Board. The initial release of data and tools used in this work was published recently [11]. Results reported are in compliance with the *All of Us* Data and Statistics Dissemination Policy disallowing disclosure of group counts under 20.

### The *All of Us* research hub

This work was performed on data collected by the previously described *All of Us* Research Program using the *All of Us* Researcher Workbench, a cloud-based platform where approved researchers can access and analyze *All of Us* data. *All of Us* data currently available include surveys, EHR, and physical measurements, with more data and data types slated to be available in future releases. The details of the surveys are available in the Survey Explorer found in the Research Hub, a website designed to support researchers [12]. Each survey includes branching logic and all questions are optional and may be skipped by the participant. Physical measurements recorded at enrollment include systolic and diastolic blood pressure, height, weight, heart rate, waist and hip measurement, wheelchair use, and current pregnancy status. EHR data was linked for those consented participants. All three datatypes (survey, physical measurements, and EHR) are mapped to the Observational Health and Medicines Outcomes Partnership (OMOP) common data model v 5.2 maintained by the Observational Health and Data Sciences Initiative collaborative. To protect participant privacy, a series of data transformations were applied. These included data suppression of codes with a high risk of identification such as military status; generalization of categories, including age, sex at birth, gender identity, sexual orientation, and race; and date shifting by a random (less than one year) number of days, implemented consistently across each participant record. Documentation on privacy implementation and creation of the curated data repository is available in the *All of Us* Registered Tier CDR Data Dictionary [13]. The Researcher Workbench currently offers tools with a user interface built for selecting groups of participants (Cohort Builder), creating datasets for analysis (Dataset Builder), and Workspaces with Jupyter Notebooks (Notebooks) to analyze data. The Notebooks enable use of saved datasets and direct query using R and Python 3 programming languages.

### Sample selection

Among *All of Us* enrollees, we selected participants 18 to 90 years of age at the time of initial enrollment, who reported either male or female sex at birth, and reported race as white, black or Asian and ethnicity as Hispanic or non-Hispanic.

## Characterization of atrial fibrillation

Presence of AF was determined through answers to self-reported questionnaires and from data obtained from EHRs. At enrollment, participants complete a series of online questionnaires. AF was considered to be present if they answered yes to the question "Has a doctor or health care provider ever told you that you have atrial fibrillation?" Participants responding yes to this question were considered to have prevalent AF.

Presence of AF in the EHR was determined if the EHR data contained two or more instances of atrial fibrillation, identified with selected Systematized Nomenclature of Medicine Clinical Terms (SNOMED CT) codes. A complete list of SNOMED CT codes and their corresponding OMOP concept IDs is provided in **S1 Table**. If the EHR-based diagnosis occurred before participant enrollment, AF was considered to be prevalent; otherwise, the diagnosis of AF was considered incident.

Of note, presence or absence of AF was not noted on the physical measurements recorded at enrollment.

## Covariates

Date of birth, sex assigned at birth, race, Hispanic ethnicity, and smoking status were self-reported using online surveys. Ever smoking was defined as having smoked over 100 cigarettes over the lifetime. Height and weight were measured by trained study technicians following a standard protocol. Systolic and diastolic blood pressure were measured three times with the participant seated. The mean of the second and third measurement was used for analysis. The following comorbidities were ascertained via self-report in surveys and in the EHR: stroke, heart failure, coronary artery disease, and diabetes.

## Statistical analysis

Participant characteristics were summarized by race/ethnicity and by participation in study components (EHR, medical history survey). Prevalence of AF was calculated as the proportion of participants with AF among all participants. We performed separate analyses among individuals participating in the medical history survey and among those whose EHR data were available. For each analysis, we calculated overall prevalence as well as sex, race/ethnicity, and age-specific prevalence. The incidence rates of newly diagnosed AF were calculated as the number of new AF diagnoses after enrollment date divided by person-years of follow-up among those without evidence of AF at enrollment and with EHR data. We calculated incidence of AF overall and by sex, race/ethnicity, and age. Association of risk factors with prevalent AF was evaluated with odds ratios (OR) and 95% confidence intervals (95% CI) from multivariable logistic regression adjusting for age, sex, and race/ethnicity in all models, while associations with incident AF were estimated with hazard ratios (HR) and 95% CI obtained from Cox regression models, also adjusting for age, sex, and race/ethnicity. Time to event was defined as the time in days between baseline and date of incident AF or August 31, 2019, whichever occurred earlier.

## Results

Between May 2017 and August 2019, there were 173,099 individuals who enrolled in the *All of Us* program and met inclusion criteria. Of these individuals, 35,483 (21%) had completed the medical history survey and 92,318 (53%) had EHR data available at the time of dataset creation. There were 20,683 participants who had both medical history survey and EHR data (12% of all eligible individuals). **Fig 1** summarizes the flow of participant selection.

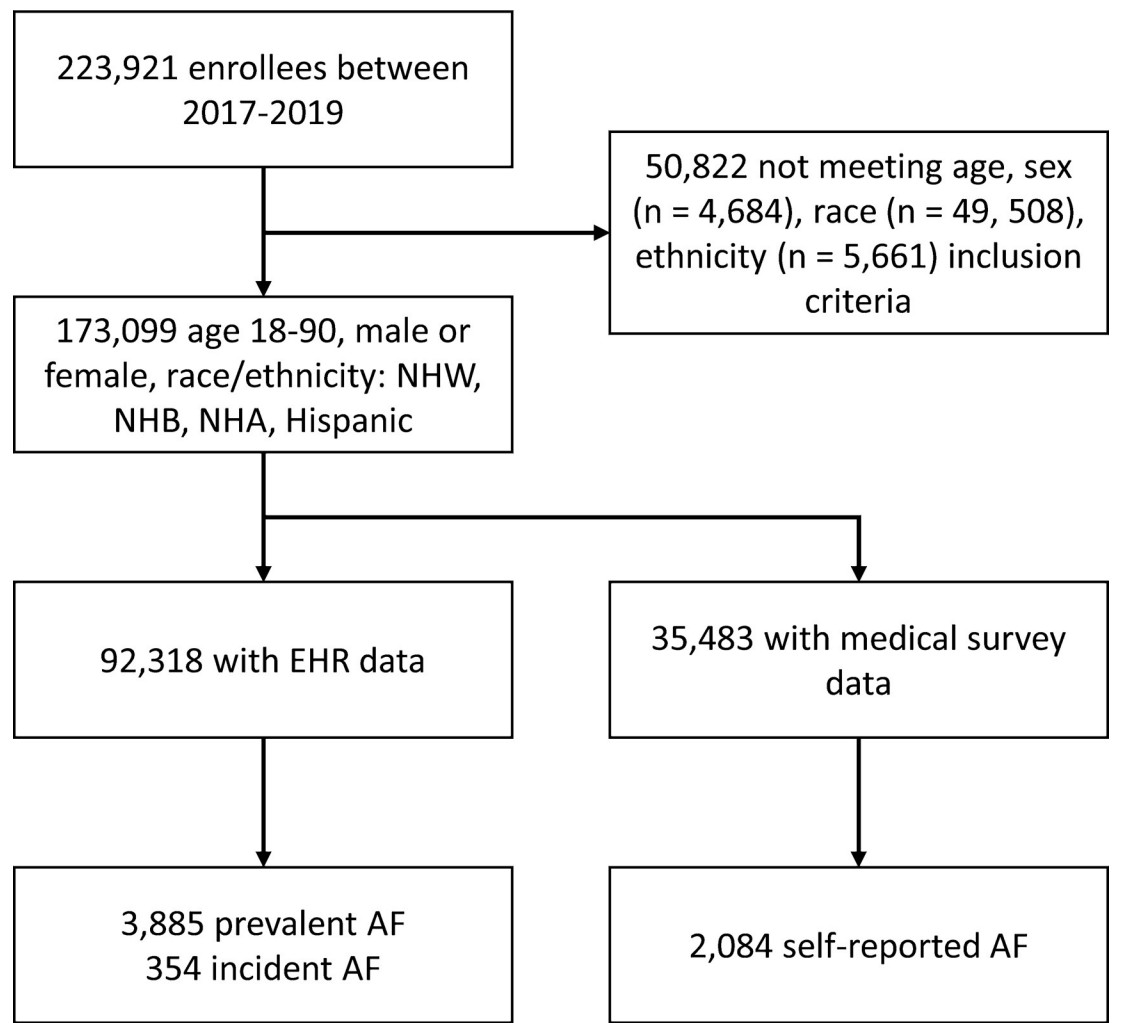

**Fig 1. Flowchart of study participants, All of Us Research Program, 2017–2019.** EHR: Electronic health records. NHA: Non-Hispanic Asian. NHB: Non-Hispanic Black. NHW: Non-Hispanic White.

Characteristics of study participants by race/ethnicity are presented in **Table 1**, while characteristics by participation in study components is provided in **S2 Table**. Approximately two thirds of participants were non-Hispanic Whites, with non-Hispanic Blacks, non-Hispanic Asians and Hispanics accounting for 26%, 4%, and 3% of participants, respectively. Overall mean age (standard deviation) was 52 (17), and females accounted for 61% of all participants. Prevalence of most cardiovascular diseases and risk factors was highest in non-Hispanic Blacks and lowest in non-Hispanic Asians (Table 1). Participants with both EHR and survey data were slightly older, more likely to be female and non-Hispanic White, and had lower prevalence of cardiovascular risk factors (smoking, diabetes) and cardiovascular diseases (stroke, heart failure, coronary heart disease) than those without data from the EHR or the medical history survey (**S2 Table**). **Fig 2** presents the overlap in participants based on availability of EHR and medical history data and diagnosis of AF. In the 20,683 participants with overlapping EHR and survey data, 1,486 had AF either from EHR or medical history survey, with 717 reported in both (48%), 208 in the EHR only (14%), and 561 in the medical survey (38%).

**Table 1. Participant characteristics by race/ethnicity.** Values correspond to mean (standard deviation) or N (%), *All of Us* Research Program, 2017–2019.

| | Non-Hispanic White | Non-Hispanic Black | Non-Hispanic Asian | Hispanic |
|---|---|---|---|---|
| N | 116,239 (67) | 44,993 (26) | 7,338 (4) | 4,529 (3) |
| Age, years* | 54 (17) | 50 (14) | 43 (17) | 40 (15) |
| Female sex | 72,129 (62) | 26,048 (58) | 4,428 (60) | 2,918 (64) |
| BMI, kg/m2 | 28.7 (6.3) | 30.3 (7.2) | 25.1 (4.7) | 29.5 (6.8) |
| SBP, mmHg | 127 (18) | 131 (20) | 122 (17) | 124 (17) |
| DBP, mmHg | 77 (11) | 81 (13) | 76 (11) | 78 (12) |
| Ever smoker | 46,390 (40) | 21,280 (47) | 1,055 (14) | 1,595 (35) |
| Diabetes | 8,773 (7.5) | 6,085 (13.5) | 381 (5.2) | 348 (7.7) |
| Heart failure | 3,034 (2.6) | 1,978 (4.4) | 74 (1.0) | 91 (2.0) |
| CHD | 6,286 (5.4) | 2,313 (5.1) | 164 (2.2) | 149 (3.3) |
| Stroke | 298 (0.3) | 175 (0.4) | <20 (<0.3) | <20 (<0.4) |

BMI: body mass index; CHD: coronary heart disease; DBP: diastolic blood pressure; SBP: systolic blood pressure.

* Age range: 18 to 90 years

Among eligible *All of Us* participants, 35,483 provided data on prior history of medical conditions, including AF. Of them, 2,084 (5.9%) reported a diagnosis of AF. Prevalence of self-reported AF by age group, race/ethnicity and sex are reported in **Table 2**. Prevalence increased with age, from <1% among participants younger than 40 to >20% in those 80 and older. Males and non-Hispanic Whites reported higher prevalence than females and other racial/ethnic groups. These differences persisted after simultaneous adjustment for age, sex and race/ethnicity, with men having approximately 70% higher odds of AF than females (OR 1.7, 95% CI 1.5, 1.8), and groups other than non-Hispanic Whites having 25–43% lower odds of self-reported AF than non-Hispanic Whites. The associations were similar when the analysis was restricted to the 20,683 participants who completed the medical history survey and had available EHR data (**S3 Table**).

A similar pattern was observed among 92,318 *All of Us* participants with linked EHR. A total of 3,885 (4.2%) participants had evidence of prevalent AF in their EHR at the time of enrollment. During a median follow-up of 16 months, an additional 354 (0.4%) participants were diagnosed with AF after enrollment. Age, sex, and race/ethnicity-specific prevalence and incidence of AF are shown in **Table 3**. Older age was strongly associated with higher prevalence of AF. For example, compared to those younger than 40, the OR of prevalent AF among those 80 and older was 66.2 (95% CI 51.5, 85.1). Males had 90% higher odds of prevalent AF than females (OR 1.9, 95% CI 1.8, 2.0), while odds of prevalent AF were lower in all race/ethnicity groups compared to non-Hispanic Whites. Similarly, older age and male sex were also associated with higher rates of incident AF. Compared to non-Hispanic Whites, both non-Hispanic Asians and non-Hispanic Blacks experienced lower rates of AF (HR 0.41, 95% CI 0.15, 1.1, and HR 0.55, 95% CI 0.40, 0.76, respectively). Rates of AF in Hispanics were higher than in non-Hispanic Whites, but this is based on a small number of AF events (<20).

**Table 4** shows the association of selected cardiovascular risk factors and cardiovascular diseases with prevalent and incident AF among participants with EHR data. Higher BMI and prior history of heart failure or coronary heart disease were associated with increased prevalence and incidence of AF. Diabetes and stroke history were associated with prevalent AF but not incident AF. Finally, neither blood pressure nor smoking status were risk factors for AF in this population. The pattern of associations was comparable among the 20,683 participants who completed the medical history survey and had available EHR data (**S3 Table**).

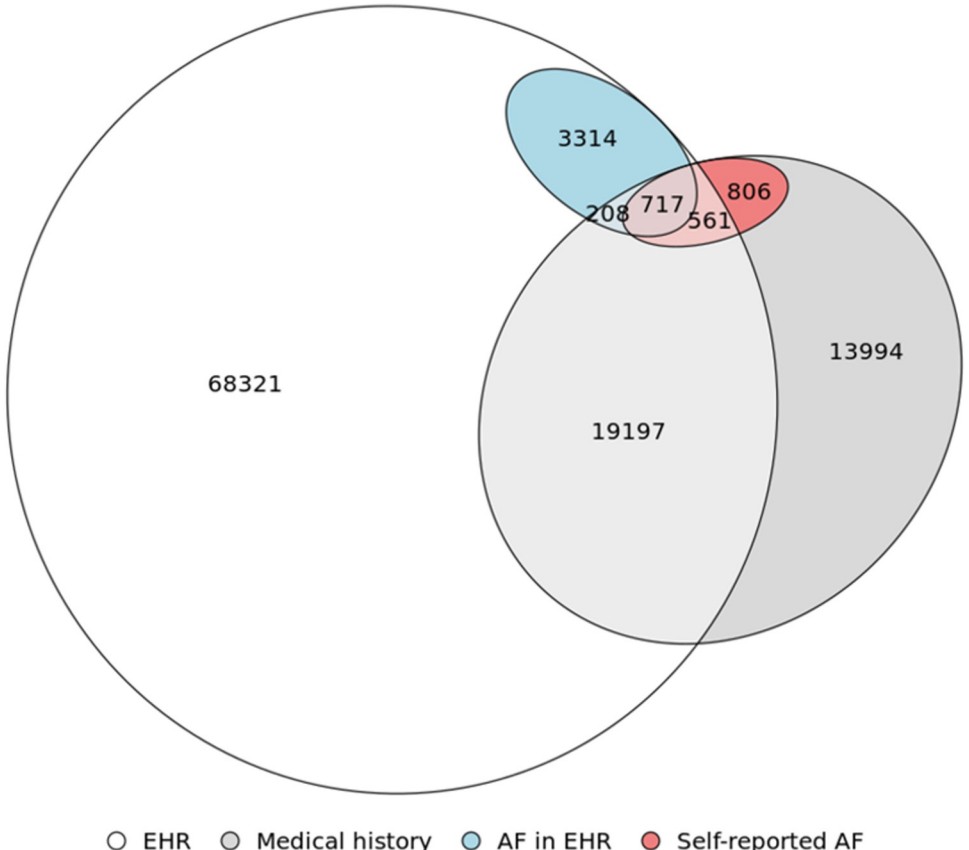

**Fig 2. Venn diagram representing overlap of participants with EHR data, medical history survey data, those with AF in the EHR, and those with self-reported AF in the medical history survey, All of Us Research Program, 2017–2019.** The intersection of these variables leads to eight different groups: (1) among participants with both EHR and medical survey, those with AF in the EHR and the medical survey (n = 717), AF in the medical survey but no in the EHR (n = 561), AF in the EHR but not in the medical survey (n = 208), and no AF in any data source (n = 19,197); (2) among participants with EHR but not medical survey, AF in the EHR (n = 3,314) and no AF (n = 68,322); and (3) among participants with medical history survey but not EHR, AF in the medical survey (n = 806) and no AF (n = 13,994). The diagram does not include 65,982 participants with neither EHR nor medical history survey.

## Discussion

Our analysis describes the epidemiologic characteristics of AF among the diverse participants in the *All of Us* Research Program. The main findings were that both prevalence and incidence of AF increased with age, were higher in males than females, and non-Hispanic Whites than in other racial/ethnic groups. These patterns are comparable to those from other epidemiologic studies in the US (**S4 and S5 Tables**). Previously described risk factors for AF, including obesity, diabetes, and prevalence of some cardiovascular diseases, were associated with the risk of AF. The overall findings from this analysis are consistent with established observations about the epidemiology of AF.

Over the last two decades, seminal studies in the epidemiology of AF conducted in community-based cohort studies have demonstrated that older age, male sex and being non-Hispanic White are strongly associated with a higher prevalence and incidence of AF [3–6]. These same studies have also contributed to identify the major risk factors for AF, including hypertension, obesity, and prior history of other cardiovascular diseases [3, 14].

Analyses of administrative databases, such as Medicare, have complemented findings from traditional cohort studies on the epidemiology of AF, offsetting their limitations in validity of

**Table 2. Prevalence of self-reported atrial fibrillation (AF) by age, sex, and race/ethnicity, *All of Us* Research Program, 2017–2019.**

| Age group | AF cases | Total | AF prevalence (%) | Odds ratio (95% CI)* |
|---|---|---|---|---|
| <40 | 80 | 8,673 | 0.9 | 1 (Ref.) |
| 40–49 | 91 | 4,213 | 2.2 | 2.3 (1.7, 3.2) |
| 50–59 | 235 | 6,416 | 3.7 | 4.0 (3.1, 5.1) |
| 60–69 | 686 | 9,193 | 7.5 | 7.9 (6.3, 10.0) |
| 70–79 | 781 | 6,029 | 13.0 | 13.8 (11.0, 17.5) |
| 80–89 | 211 | 959 | 22.0 | 25.8 (19.7, 33.9) |
| Sex | | | | |
| Female | 1,014 | 23,326 | 4.3 | 1 (Ref.) |
| Male | 1,070 | 12,157 | 8.8 | 1.7 (1.5, 1.8) |
| Race / ethnicity | | | | |
| Hispanic | <20 | 743 | <2.7 | 0.75 (0.47, 1.2) |
| NH Asian | 27 | 1,295 | 2.1 | 0.57 (0.38, 0.84) |
| NH Black | 63 | 2,239 | 2.8 | 0.61 (0.47, 0.79) |
| NH White | 1,976 | 31,206 | 6.3 | 1 (Ref.) |

* Results from binary logistic regression including age category, sex and race as independent variables in the same model.

disease and risk factor phenotyping by their large sample size and broader geographic distribution [8, 9]. In the United States, however, these administrative databases are limited to certain age groups (Medicare beneficiaries older than 65), individuals from specific socioeconomic strata (Medicaid beneficiaries), types of healthcare facilities, or geographic area (state-based databases). Nonetheless, their findings are consistent with those of traditional cohort studies [3–5, 14].

In comparison with other studies, the *All of Us* Research Program provides a unique opportunity to characterize the epidemiology and risk factors of common chronic conditions,

**Table 3. Prevalence and incidence of EHR-derived AF by age, sex, and race/ethnicity, *All of Us* Research Program, 2017–2019.**

| | Prevalent AF | | | | Incident AF | | | |
|---|---|---|---|---|---|---|---|---|
| Age group | AF cases | Total | Prevalence (%) | Odds Ratio (95% CI)* | AF cases | Person-years | Incidence** | Hazard Ratio (95% CI)*** |
| <40 | 71 | 23,285 | 0.3 | 1 (Ref.) | <20 | 32,271 | <0.6 | 1 (Ref.) |
| 40–49 | 160 | 12,609 | 1.3 | 4.2 (3.2, 5.6) | <20 | 17,006 | <1.2 | 2.9 (1.4, 5.9) |
| 50–59 | 466 | 19,326 | 2.4 | 8.0 (6.2, 10.3) | 57 | 25,830 | 2.2 | 6.0 (3.2, 11.2) |
| 60–69 | 1150 | 21,016 | 5.5 | 17.8 (14.0, 22.6) | 110 | 26,893 | 4.1 | 10.4 (5.7, 19.0) |
| 70–79 | 1445 | 12,979 | 11.1 | 35.8 (28.1, 45.5) | 116 | 15,314 | 7.6 | 17.4 (9.6, 31.7) |
| 80–89 | 593 | 3,103 | 19.1 | 66.2 (51.5, 85.1) | 41 | 3,232 | 12.3 | 28.1 (14.7, 53.7) |
| Sex | | | | | | | | |
| Female | 1,751 | 57,439 | 3.0 | 1 (Ref.) | 170 | 76,083 | 2.2 | 1 (Ref.) |
| Male | 2,134 | 34,879 | 6.1 | 1.9 (1.8, 2.0) | 184 | 44,464 | 4.1 | 1.7 (1.4, 2.1) |
| Race / ethnicity | | | | | | | | |
| Hispanic | 46 | 2,496 | 1.8 | 0.89 (0.66, 1.2) | <20 | 3,372 | <5.9 | 1.4 (0.71, 2.9) |
| NH Asian | 62 | 3,444 | 1.8 | 0.63 (0.49, 0.82) | <20 | 4,850 | <4.1 | 0.41 (0.15, 1.1) |
| NH Black | 544 | 24,435 | 2.2 | 0.64 (0.58, 0.71) | 48 | 31,102 | 1.5 | 0.55 (0.40, 0.76) |
| NH White | 3,233 | 61,943 | 5.2 | 1 (Ref.) | 294 | 81,222 | 3.6 | 1 (Ref.) |

* Results from binary logistic regression including age category, sex and race as independent variables in the same model.

** Per 1,000 person-years.

*** Results from Cox regression including age category, sex and race as independent variables in the same model.

**Table 4. Multivariable model for prevalent and incident AF (odds ratio or hazard ratio for covariates),** *All of Us* **Research Program 2017–2019.**

|  | Prevalent AF | Incident AF |
| --- | --- | --- |
|  | Odds ratio (95% CI)* | Hazard ratio (95% CI)** |
| BMI, per 5 kg/m² | 1.1 (1.1, 1.2) | 1.2 (1.1, 1.3) |
| SBP, per 20 mmHg | 0.78 (0.74, 0.82) | 0.98 (0.84, 1.1) |
| DBP, per 10 mmHg | 1.1 (1.1, 1.20) | 1.0 (0.93, 1.2) |
| Ever smoking | 0.93 (0.86, 1.0) | 1.1 (0.86, 1.3) |
| Diabetes | 1.2 (1.1, 1.3) | 1.0 (0.77, 1.3) |
| Heart failure | 6.6 (6.0, 7.2) | 3.4 (2.5, 4.7) |
| Coronary heart disease | 2.7 (2.5, 2.9) | 1.6 (1.2, 2.1) |
| Stroke | 2.9 (2.2, 3.8) | 1.2 (0.39, 3.8) |

* Binary logistic regression adjusted for age, sex, race/ethnicity and all the variables included in the table.

** Cox regression adjusted for age, sex, race/ethnicity and all the variables included in the table.

including AF, in a large and extremely diverse population. Given its specific focus on recruiting groups previously underrepresented in biomedical research, findings from *All of Us* have the potential of being more generalizable to the overall population. In this context, our analysis has replicated previously described aspects of the epidemiology of AF. As the *All of Us* Research Program develops and other types of data become available, including genomic information, the rich demographic and geographic diversity among participants will enhance research among underrepresented groups. For example, the most recent published genome-wide association study of AF included less than 9,000 Blacks, of which 1,307 had AF, and approximately 3,000 Hispanics (277 AF cases) [15]. The *All of Us* Research Program will eventually provide data orders of magnitude larger, making possible to understand the influence of ancestry on AF risk and the impact that socioeconomic adversity, disproportionally affecting some underrepresented groups, has on AF risk and outcomes [16, 17].

Most of the explored associations were consistent with previous literature. Older age, male sex and non-Hispanic white race, prevalent diabetes, a prior history of cardiovascular disease (heart failure, stroke, coronary artery disease), as well as higher body mass index, were all associated with higher risk of AF. Most of these risk factors are part of established scores for the prediction of AF [14]. However, elevated systolic blood pressure, which is a consistently described risk factor for AF, was not associated with higher AF risk in the *All of Us* sample. This may be due to not accounting for the use of antihypertensive medication.

We draw two major conclusions from these results. First, by observing the expected associations of demographic and clinical factors with the risk of AF, we are indirectly supporting the validity of the *All of Us* Research Program data for future studies on the epidemiology of AF and its outcomes. Second, our findings provide additional evidence suggesting that AF rates are highest among non-Hispanic Whites. Ancestry-related genetic factors seem to be partly responsible for this, as shown by higher risk of AF associated with increased degrees of European ancestry among African Americans [18]. Though a few reports suggest that under-ascertainment among non-Whites could partially explain these differences [19], most studies using extended electrocardiographic recordings or pacemaker data and, therefore, less likely to have biased ascertainment, confirm these racial/ethnic differences [20–23].

Our analysis has some major strengths, including the large sample size and the racial/ethnic diversity of the study population. Absence of validated AF diagnoses, lack of AF ascertainment at the time of enrollment physical measurements, and missing EHR or medical survey data in

large subsets of the study population can be considered major weaknesses. Previous studies, however, have demonstrated the validity of administrative diagnostic data to identify AF diagnoses [24]. Also, using medical history or EHR data resulted in comparable patterns of AF prevalence by age, sex and race/ethnicity, indicating that both sources of information may offer valid data.

In conclusion, in a novel research resource such as the *All of Us* Research Program, we identified epidemiologic patterns of AF similar to what studies with different designs and methods have reported. These findings provide indirect evidence of the validity of data collected by the *All of Us* Research Program and support the value of this resource to conduct studies on the epidemiology of AF and, potentially, other cardiovascular diseases.

## Supporting information

**S1 Table. Concept IDs used to identify atrial fibrillation in the electronic health records, *All of Us* Research Program, 2017–2019.**
(DOCX)

**S2 Table. Participant characteristics by study component participation.** Values correspond to mean (standard deviation) or N (%), *All of Us* Research Program 2017–2019.
(DOCX)

**S3 Table. Association of age, sex, race/ethnicity and cardiovascular risk factors with prevalent atrial fibrillation among *All of Us* participants with EHR and medical history survey data (N = 20,683), *All of Us* Research Program, 2017–2019.**
(DOCX)

**S4 Table. Age, sex, and race/ethnicity prevalence of AF in the *All of Us* Research Program and selected epidemiologic studies in the United States.** Values correspond to prevalence per 100 persons.
(DOCX)

**S5 Table. Age, sex, and race/ethnicity incidence rates of AF in the *All of Us* Research Program and selected epidemiologic studies in the United States.** Values correspond to incidence rate per 1000 person-years.
(DOCX)

## Acknowledgments

The *All of Us* Research Program would not be possible without the partnership of its participants. We are also grateful by the contributions of *All of Us* Research Program investigators. Dr. Gebo initiated this work while serving as Chief Medical and Scientific Officer of the *All of Us* Research Program.

## Author Contributions

**Conceptualization:** Alvaro Alonso.

**Formal analysis:** Alvaro Alonso.

**Methodology:** Alvaro Alonso, Hooman Kamel, Vignesh Subbian, Jun Qian, Eric Boerwinkle, Mine Cicek, Cheryl R. Clark, Elizabeth G. Cohn, Kelly A. Gebo, Roxana Loperena-Cortes, Kelsey R. Mayo, Stephen Mockrin, Lucila Ohno-Machado, Sheri D. Schully, Andrea H. Ramirez, Philip Greenland.

**Writing – original draft:** Alvaro Alonso.

**Writing – review & editing:** Alvaro Alonso, Aniqa B. Alam, Hooman Kamel, Vignesh Subbian, Jun Qian, Eric Boerwinkle, Mine Cicek, Cheryl R. Clark, Elizabeth G. Cohn, Kelly A. Gebo, Roxana Loperena-Cortes, Kelsey R. Mayo, Stephen Mockrin, Lucila Ohno-Machado, Sheri D. Schully, Andrea H. Ramirez, Philip Greenland.

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
