## [Decision Letter · Decision Letter 0]

13 Dec 2021

PONE-D-21-37941Epidemiology of atrial fibrillation in the All of Us Research ProgramPLOS ONE

Dear Dr. Alonso,

Thank you for submitting your manuscript to PLOS ONE. After careful consideration, we feel that it has merit but does not fully meet PLOS ONE’s publication criteria as it currently stands. Therefore, we invite you to submit a revised version of the manuscript that addresses the points raised during the review process.

We look forward to receiving your revised manuscript.

Kind regards,

Prof. Gaetano Santulli, MD

Academic Editor

PLOS ONE

Journal Requirements:

Reviewers' comments:

Reviewer's Responses to Questions

**Comments to the Author**

1. Is the manuscript technically sound, and do the data support the conclusions?

Reviewer #1: Yes

Reviewer #2: Partly

2. Has the statistical analysis been performed appropriately and rigorously? 

Reviewer #1: Yes

Reviewer #2: N/A

3. Have the authors made all data underlying the findings in their manuscript fully available?

Reviewer #1: Yes

Reviewer #2: Yes

4. Is the manuscript presented in an intelligible fashion and written in standard English?

Reviewer #1: Yes

Reviewer #2: Yes

5. Review Comments to the Author

Reviewer #1: This study proposes to establish the prevalence, incidence, and risk factors for atrial fibrillation in a large cohort.

Among 92,318 participants with available EHR data, 3,885 (4.2%) had AF at the time of study enrolment. During a median follow-up of 16 months, 354 new cases of AF were identified among 88,433 eligible participants.

This study confirms information already known, but remains interesting. Its main problem concerns recruitment bias and the reliability of the information collected. Indeed, AF was not noted on the physical measurements recorded at enrollment. This would have avoided some false positives and false negatives. Indeed, the way AF is diagnosed, either at enrolment or in the follow-up, inevitably leads to diagnostic errors.

Major concern

1/ You need to be clearer about the number of patients evaluated, throughout the paper.

For example, 35,483 (21%) had completed the medical history survey and 92,318 (53%) had EHR data available at the time of dataset creation. There were 20,683 participants who had both medical history survey and EHR data. So your data is from how many unique patients actually? 107,118 ?

2/ Can you detail the number of 50,822 excluded patients for not meeting inclusion criteria ? This seems like a high number when you start with 223,921 patients. How many because of age? gender? race?

3/ The result on blood pressure is very surprising. It needs to be better explained in the discussion.

Minor concerns

1/ Figure 1 should be reworked according to the usual presentation of a flow chart.

2/ I don't understand why you give the total population statistics in table 1, when you have usable data for only a portion of them in the rest of the study.

Reviewer #2: The manuscript aimed to investigate the prevalence, incidence and risk factors of atrial fibrillation in a large, geographically and ethnically diverse cohort in the United States.

The idea is interesting with few elements of novelty. The sample size is adequate but the study suffers from a lack of iconography and the statistical analysis is poor. The discussion and the reference section needs to be improved. Please cite and discuss the role of comorbidities in atrial fibrillation:

- 34793855

- 28450367

- 34440883

- 33238738

- 34101155

- 30926014

- 32495723

6. PLOS authors have the option to publish the peer review history of their article (what does this mean?). If published, this will include your full peer review and any attached files.

Reviewer #1: No

Reviewer #2: No

---

## [Editor Report · Decision Letter 1]

3 Mar 2022

Epidemiology of atrial fibrillation in the All of Us Research Program

PONE-D-21-37941R1

Dear Dr. Alonso,

We’re pleased to inform you that your manuscript has been judged scientifically suitable for publication and will be formally accepted for publication once it meets all outstanding technical requirements.

Kind regards,

Gaetano Santulli, MD

Academic Editor

PLOS ONE
---

## [Editor Report · Acceptance letter]

7 Mar 2022

PONE-D-21-37941R1 

Epidemiology of atrial fibrillation in the *All of Us* Research Program 

Dear Dr. Alonso:

I'm pleased to inform you that your manuscript has been deemed suitable for publication in PLOS ONE. Congratulations! Your manuscript is now with our production department. 

Kind regards, 

on behalf of

Professor Gaetano Santulli 

Academic Editor

PLOS ONE